# CERTIFYING NEURAL NETWORK AUDIO CLASSIFIERS

## ABSTRACT

We present the first end-to-end verifier of audio classifiers. Compared to existing methods, our approach enables analysis of both, the entire audio processing stage as well as recurrent neural network architectures (e.g., LSTM). The audio processing is verified using novel convex relaxations tailored to feature extraction operations used in audio (e.g., Fast Fourier Transform) while recurrent architectures are certified via a novel binary relaxation for the recurrent unit update. We show the verifier scales to large networks while computing significantly tighter bounds than existing methods for common audio classification benchmarks: on the challenging Google Speech Commands dataset we certify 95% more inputs than the interval approximation (only prior scalable method), for a perturbation of -90dB.

## 1 INTRODUCTION

Recent advances in deep learning have enabled replacement of traditional voice recognition systems with a single neural network trained from data (Graves et al., 2013; Hannun et al., 2014; Amodei et al., 2016). Wide adoption of these networks in consumer devices poses a threat to their safety when exposed to a malicious adversary. Indeed, it was recently shown that an adversary can inject noise unrecognizable to a human and force the network to misclassify (Szegedy et al., 2013; Goodfellow et al., 2014; Zhang et al., 2017; Carlini & Wagner, 2018; Carlini et al., 2016; Qin et al., 2019; Neekhara et al., 2019; Yang et al., 2019; Esmaeilpour et al., 2019), exposing a serious security flaw.

Ideally, when deploying an automated speech recognition system we would like to guarantee that the system is robust against noise injected by an adversary. There has been substantial recent work on certifying robustness of computer vision models (Katz et al., 2017; Ehlers, 2017; Bunel et al., 2018; Ruan et al., 2018; Tjeng et al., 2019; Anderson et al., 2018; Wong et al., 2018; Dvijotham et al., 2018; Raghunathan et al., 2018; Dvijotham et al., 2019; Weng et al., 2018; Zhang et al., 2018; Salman et al., 2019; Gehr et al., 2018; Singh et al., 2018; 2019a; Wang et al., 2018; Singh et al., 2019b). However, the audio domain poses unique challenges not addressed by prior certification work for vision.

**Differences between audio and vision models** Concretely, while an input to a vision model is a raw image, audio models typically come with a complex preprocessing stage (that involves non-trivial non-linear operations such as logarithm) which extracts relevant features from the signal. Additionally, audio systems typically use recurrent architectures (Chiu et al., 2017) which computer vision verifiers do not handle as they focus on fully-connected, convolutional and residual architectures.

**This work** We address both of these challenges and propose an end-to-end verification method for neural network based audio classifiers and an implementation of this method in a system called DAC (Deep Audio Certifier). Our threat model assumes an attacker can introduce a noise-based perturbation to the raw audio input signal. The goal then is to certify that, for any signal that the attacker can produce, the neural network classifies the signal to the correct label. We perform verification of this property using the framework of abstract interpretation (Gehr et al., 2018). At a high level, the idea is to maintain an abstraction capturing all possible behaviors of both the audio processing stage and the neural network. The flow of DAC is shown in Fig. 1 where all abstractions are dark blue shapes.

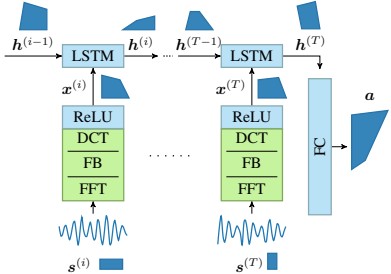

Figure 1: End-to-End Audio Certification Flow using DAC.

Here, all possible signals an attacker can obtain are captured using an abstraction $s^{(i)}$ (a convex relaxation). This abstraction is then propagated through the audio processing stage (shown in green boxes). The key components of this step are *abstract transformers*. For each audio processing operation (e.g. FFT) we create an abstract transformer which receives an abstraction representing an approximation of all possible inputs to the operation and outputs a new abstraction which approximates all possible outputs of the operation. The result of the audio processing stage is the abstraction $x^{(i)}$.

The shape $x^{(i)}$ is then used as input to the recurrent LSTM unit (light blue) which maintains an abstraction of a hidden state $h^{(i-1)}$. LSTM consists of multiple operations and we create a custom abstract transformer for each of those. The result of the transformers in LSTM is a new hidden state $h^{(i)}$. If this was the last frame in the signal (meaning $i = T$), then hidden state $h^{(T)}$ is passed through the fully connected layer of the neural network and, again using the abstract transformer, the final abstract shape $a$ is obtained at the output (at the right of Fig. 1). Finally, to certify the property we check if each concrete output in the abstraction $a$ classifies to the correct label (this is typically easy). If this is true, the output of the network is correct for all inputs that the attacker can create.

**Related work on RNN certification** The work of (Ko et al., 2019) proposes the POPQORN verifier for recurrent neural networks (RNN). We note that POPQORN does not handle the audio preprocessing pipeline. Even though POPQORN cannot directly verify audio classifiers, their approximations for LSTM non-linearities can be integrated in DAC. This results in $\approx 200\times$ slowdown with small decrease in the volume of the approximation. The massive slowdown makes their approximations unsuitable for certifying audio classifiers. In contrast, using our custom abstract transformers for LSTM non-linearities, DAC can precisely certify end-to-end robustness of challenging audio classifiers in few minutes.

Our main contributions are:

1. A novel and efficient method to certify robustness of neural network audio classifiers to noise-based perturbations. The method is based on new abstract transformers which handle non-linear operations used in both audio processing and recurrent architectures.

2. An implementation of both verification and provably robust training in a system called DAC. We evaluated DAC on common audio classification benchmarks, showing it scales to realistic networks and is far more precise (97% to 2%) than the next best scalable method.

## 2 BACKGROUND

We first define a threat model that we work with and then present all operations that are part of the verification procedure, including audio processing (MFCC) and LSTM updates. We also discuss the type of verification method we employ.

**Threat model** We follow the same attacker threat model as Carlini & Wagner (2018). The assumption is that the attacker can add noise $\delta$ to the original signal $s$ so to obtain a perturbed signal $s' = s + \delta$. The measure of signal distortion are decibels (dB) defined as:

$$dB(s) = \max_i 20 \cdot \log_{10}(|s_i|); \; dB_s(\delta) = dB(\delta) - dB(s)$$

Note that the quieter the noise is, the smaller the decibel of perturbation $dB_s(\delta)$ (it is usually a negative value as it is quieter than the signal). We assume the attacker can generate noise $\delta$ such that $dB_s(\delta) < \epsilon$ where $\epsilon$ is a constant defined by the threat model. Our goal is to verify whether the neural network classifies $s'$ correctly for each small perturbation $\delta$ (as constrained above).

**Mel-Frequency Cepstral Coefficients (MFCC)** Though there have been number of works which operate directly on the raw signal (Pascual et al., 2017; Sainath et al., 2015), *Mel-Frequency Cepstrum* (MFC) is traditionally preferred for audio preprocessing in speech recognition systems e.g., Deep-Speech (Hannun et al., 2014). The idea of MFC is to model non-linear human acoustic perception as power spectrum filters based on certain frequencies, called *Mel-frequencies*. The final result of the transformation is a vector of coefficients whose elements contain log-scaled values of filtered spectra, one for every Mel-frequency. This resulting vector is a feature representation of the original signal and can now be used in a downstream task such as audio classification.

Sahidullah & Saha (2012) presented an approach to represent MFCC computation using several matrix operations which we integrate with our verification framework. Given $T$ frames of audio signals of

the length $N = 2^k$, represented as matrix $\boldsymbol{S} = [\boldsymbol{s}^{(1)} \cdots \boldsymbol{s}^{(T)}]^{tr} \in \mathbb{R}^{T \times N}$ ($tr$ for transpose), audio preprocessing with MFCC is calculated using the following steps:

1. *Pre-emphasizing and Windowing:* $\boldsymbol{Y} = \boldsymbol{S}(\boldsymbol{I}_N - c_{pe}\boldsymbol{I}_N^{+1}) \odot \boldsymbol{H}$
   Transform the signal with the pre-emphasizing and applying the Hamming window. Here, $\boldsymbol{I}_N^{+1} \in \mathbb{R}^{N \times N}$ is the shifted diagonal identity matrix ($I_{i,j}^{+1} = 1$ if $i + 1 = j$, otherwise 0), $\boldsymbol{H} \in \mathbb{R}^{T \times N}$ is the Hamming window, and $c_{pe}$ is the pre-emphasizing constant.

2. *Power Spectrum of Fast Fourier Transform (FFT):* $\boldsymbol{\Theta} = (\boldsymbol{YW}) \odot (\boldsymbol{YW})$
   Perform FFT on the windowed data and square it to get the real-value spectrum. We can denote FFT on discrete domain (DFT) with the multiplication of $\boldsymbol{Y}$ and $\boldsymbol{W} \in \mathbb{C}^{N \times N/2}$.

3. *Filter Bank Log Energy:* $\boldsymbol{\Psi} = \log(\boldsymbol{\Theta \Lambda})$
   Apply the Mel frequency filter bank to the power spectrum and get the log value of them. $\boldsymbol{\Lambda} \in \mathbb{R}^{N/2 \times p}$ is the filter bank given the number of filters $p$, and $\log$ is applied entry-wise.

4. *DCT(Discrete Cosine Transformation):* $\boldsymbol{X} = \boldsymbol{\Psi D}$
   Perform DCT on the previous result. Again, this can be formulated using matrix multiplication. We use the resulting $\boldsymbol{X} = [\boldsymbol{x}^{(1)} \cdots \boldsymbol{x}^{(T)}]^{tr}$ as the input for the neural network.

**Long-Short Term Memory (LSTM)** LSTM architectures (Hochreiter & Schmidhuber, 1997) are a key part in modern state-of-the-art speech recognition systems (Hannun et al., 2014; Amodei et al., 2016). In our work, we consider the following definitions of updates in the LSTM unit.

$$\boldsymbol{f}_0^{(t)} = [\boldsymbol{x}^{(t)}, \boldsymbol{h}^{(t-1)}]\boldsymbol{W}_f + \boldsymbol{b}_f \qquad\qquad \boldsymbol{o}_0^{(t)} = [\boldsymbol{x}^{(t)}, \boldsymbol{h}^{(t-1)}]\boldsymbol{W}_o + \boldsymbol{b}_o$$

$$\boldsymbol{i}_0^{(t)} = [\boldsymbol{x}^{(t)}, \boldsymbol{h}^{(t-1)}]\boldsymbol{W}_i + \boldsymbol{b}_i \qquad\qquad \tilde{\boldsymbol{c}}_0^{(t)} = [\boldsymbol{x}^{(t)}, \boldsymbol{h}^{(t-1)}]\boldsymbol{W}_{\tilde{c}} + \boldsymbol{b}_{\tilde{c}}$$

$$\boldsymbol{c}^{(t)} = \sigma(\boldsymbol{f}_0^{(t)}) \odot \boldsymbol{c}^{(t-1)} + \sigma(\boldsymbol{i}_0^{(t)}) \odot \tanh(\tilde{\boldsymbol{c}}_0^{(t)}) \qquad \boldsymbol{h}^{(t)} = \sigma(\boldsymbol{o}_0^{(t)}) \odot \tanh(\boldsymbol{c}^{(t)})$$

where $[\cdot, \cdot]$ is the horizontal concatenation two row vectors, $\boldsymbol{W}.$ and $\boldsymbol{b}.$ are kernel and bias of the cell, respectively. At timestep $t$, vectors $\boldsymbol{f}_0^{(t)}, \boldsymbol{i}_0^{(t)}, \boldsymbol{o}_0^{(t)}, \tilde{\boldsymbol{c}}_0^{(t)}$ represent pre-activations of the forget, input, and output gate, respectively, and the pre-calculation of cell state. Cell state $\boldsymbol{c}^{(t)}$ and hidden state $\boldsymbol{h}^{(t)}$ computed at timestep $t$ are propagated to the LSTM unit at the next timestep, thus allowing it to maintain states. This recurrent architecture allows inputs with arbitrary length, making it especially suited for audio inputs.

**Robustness certification** In this work, our goal will be to certify the robustness of an audio classification pipeline (including the LSTM) to noise perturbations of the input. To build such a verifier, we leverage the general method of abstract interpretation suggested by Cousot & Cousot (1977), successfully employed by some of the most recent state-of-the-art verifiers of neural networks (Gehr et al., 2018; Singh et al., 2019a). The basic idea here will be to propagate the possible perturbations (captured by a convex region) through the operations of the entire audio pipeline and to then use the final output region to certify the robustness property. The most challenging step of this approach is defining efficient yet precise approximations of the non-linear operations (called abstract transformers) used in the audio pipeline. In this work, we will introduce a number of such new abstract transformers that handle the non-linear operations used in audio processing. Specifically, our over-approximations will be expressed in the recent DEEPPOLY abstraction (Singh et al., 2019a) (a restricted form of convex polyhedra) which aims to balance efficiency and precision. That is, the abstract transformer of a non-linear function will take as input a convex polyhedra element expressible in DEEPPOLY and output another polyhedra which over-approximates the behavior of the non-linear operation when invoked with any concrete point inside the original polyhedra.

## 3 OVERVIEW OF VERIFICATION PROCESS

We now explain the workings of our verifier on a (toy) example from the point where sound enters the processing stage to where the output of the classifier is certified as robust under any allowed perturbation of the threat model. Our goal is to provide an intuitive understanding, formal details are provided later.

**Audio preprocessing** Unlike in standard computer vision tasks where the input is fed directly to the neural network, audio models typically first perform MFCC preprocessing to extract useful features

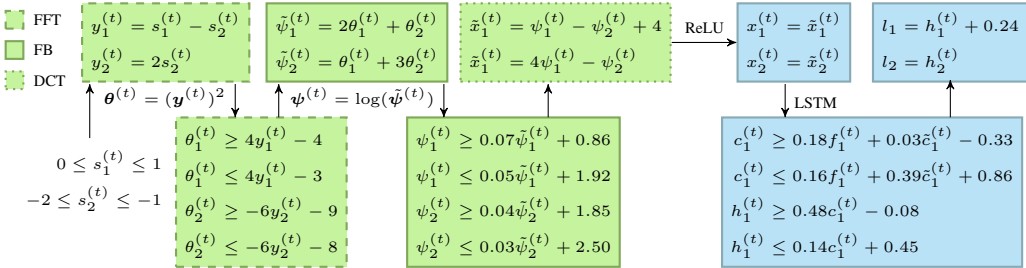

Figure 2: Robustness verification illustrated on a toy example. Each box shows the constraints computed by the verifier after processing a given operation (shown on edges). Green color denotes elements of the MFCC audio processing stage: the Fast Fourier Transform (FFT), the Filterbank Transform (FB), and the Discrete Cosine Transform (DCT). Blue color denotes components of the network: ReLU layer, LSTM and Fully Connected layer. Approximations computed with this work are shown in the bottom row. The operations which the abstract transformers handle are shown next to the downward edges. Edges without labels represent affine transforms.

from the signal. We represent all preprocessing operations as green boxes in Fig. 2. The calculation follows steps described formally in Section 2, using the same notation for resulting vectors.

**Fast Fourier transform** The first operation is Fast Fourier Transform (FFT) of the pre-emphasized input signal. It is a two-step process shown in dashed boxes in Fig. 2. We decompose it into affine and square operations which transform signal $s^{(t)}$ into an intermediate representation $\theta^{(t)}$. Using our novel and optimal abstract transformer for the square operation, formally defined in Section 4 and visualized in Fig. 3b, we obtain linear bounds on $\theta^{(t)}$.

**Filterbank transform** The next operation is Filterbank transform (FB) which consists of affine and logarithm operations, shown in solid boxes in Fig. 2. Note that if the approximations from the square transformer allows negative values, then the entire analysis will fail as the logarithm operation is undefined for negative inputs. Our transformers are carefully designed to avoid this scenario. To obtain bounds on the output of the logarithm we apply our novel and optimal logarithm transformer, also formally described in Section 4 and visualized in Fig. 3a, and obtain linear upper and lower bounds on the log energy of the filtered power spectrum, $\psi^{(t)}$. Logarithm operation is followed by Discrete Cosine Transform (DCT) resulting in a vector $\tilde{x}^{(t)}$ which is then used as an input to the fully connected layer of the neural network followed by a ReLU. Our analysis (detailed calculation in Appendix C) produces $\tilde{x}_1^{(t)} \in [0.87, 5.56], \tilde{x}_2^{(t)} \in [0.17, 12.8]$. Since all the values are positive, the following ReLU has no effect on its input and thus we set $x^{(t)} = \tilde{x}^{(t)}$. Using the back-substitution technique we describe later, we derive $x_1^{(t)} \in [0.87, 5.56], \; x_2^{(t)} \in [0.17, 12.83]$. In the next paragraph we describe bound propagation through LSTM in more detail.

**LSTM bound propagation** Here we provide a technical overview of our LSTM transformer, formalized in Section 5 and visualized in Fig. 4, by calculating the result of the transformation on the first neuron of the LSTM hidden layer. We provide detailed mathematical basis of this process in Appendix D. For our toy example, let the gates be updated as $f_1^{(t)} = x_1^{(t)}/4$, input gate $i_1^{(t)} = 2h_2^{(t-1)}$, output gate $o_1^{(t)} = h_1^{(t-1)}$ and cell state $\tilde{c}^{(t)} = h_1^{(t-1)} - x_2^{(t)}/4$. Also, assume the previous states are bounded by: $h_1^{(t-1)}, h_2^{(t-1)}, c_1^{(t-1)}, c_2^{(t-1)} \in [0.90, 1.00]$. We now apply our $\sigma(x) \cdot \tanh(y)$ and $\sigma(x) \cdot y$ transformers to get bounds of the cell state $c_1^{(t)} = \sigma(f_1^{(t)}) \cdot c_1^{(t-1)} + \sigma(i_1^{(t)}) \cdot \tanh(\tilde{c}_1^{(t)})$.

$$0.18f_1^{(t)} + 0.45 \le \sigma(f_1^{(t)}) \cdot c_1^{(t-1)} \le 0.16f_1^{(t)} + 0.58$$
$$0.03\tilde{c}_1^{(t)} - 0.78 \le \sigma(i_1^{(t)}) \cdot \tanh(\tilde{c}_1^{(t)}) \le 0.39\tilde{c}_1^{(t)} + 0.28$$

Summing up the inequalities above results in $c_1^{(t)} \in [-0.36, 1.46]$. For the hidden state $h_1^{(t)} = \sigma(o_1^{(t)}) \cdot \tanh(c_1^{(t)})$, we again apply our abstract transformer and obtain

$$0.48c_1^{(t)} - 0.08 \le \sigma(o_1^{(t)}) \cdot \tanh(c_1^{(t)}) \le 0.14c_1^{(t)} + 0.45$$

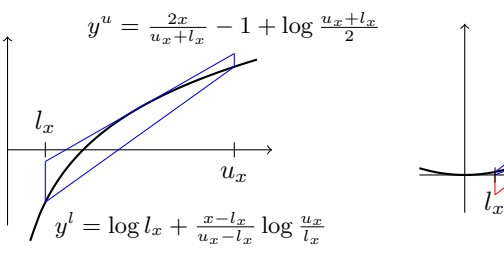
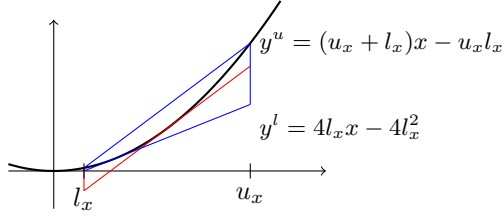

(a) Log abstract transformer.          (b) Square abstract transformer.

Figure 3: Our DeepPoly approximations of (a) the natural logarithm and (b) the square function. Blue lines represent the valid (in case of square, non-negative) upper and lower bounds which minimize the area between the planes under the given domain $[l_x, u_x]$. The red bound on (b) grants smaller area, but contains the negative value, which occurs the analysis failure in the audio processing pipeline.

i.e., $h_1^{(t)} \in [-0.25, 0.65]$. An analogous computation is performed for $h_2^{(t)}$. For this example, we assume the computation results in $h_2^{(t)} = 0$.

**Robustness certification using DeepPoly** The hidden states in the LSTM at the final timestep are passed through a fully connected layer without activation. In our example we assume that logits after the layer are obtained with the following formula: $l_1 = h_1^{(t)} + 0.24$ and $l_2 = h_2^{(t)}$. These are shown in the final box of Fig. 2. To certify that the neural network classifies the input to class 1, we need to prove that $l_1 - l_2 > 0$. We now apply the back-substitution technique as in Singh et al. (2019a) upto the input to the LSTM:

$$l_1 - l_2 = h_1^{(t)} - h_2^{(t)} + 0.24 \geq 0.48(0.18 f_1^{(t)} + 0.03\tilde{c}_1^{(t)} - 0.33) - 0.08 + 0.24$$
$$= 0.09 f_1^{(t)} + 0.01\tilde{c}_1^{(t)} + 0.01 = 0.09(x_1^{(t)}/4) + 0.01(h_1^{(t-1)} - x_2^{(t)}/4) + 0.01 \geq 0.006$$

As this value is greater than 0, robustness is established. The process above, of replacing a variable with its constraints, is called *back-substitution*. Here, we replaced $l_1$ with $h_1^{(t)} + 0.24$, then $h_1^{(t)}$ with $0.18 f_1^{(t)} + 0.03\tilde{c}_1^{(t)} - 0.33$ and so on. In Appendix C we show more detailed calculations to obtain tighter bound 0.0375 by also back-substituting in the preprocessing pipeline. For our experiments in Section 6, we tune the number of back-substitution steps to achieve a good tradeoff between speed and precision. Note that robustness cannot be proved if one concretizes the expression above to an interval, instead of performing back-substitution of linear bounds. For instance, if we concretize $h_1^{(t)}$ to $[-0.25, 0.65]$, we obtain $l_1 - l_2 \geq h_1^{(t)} - h_2^{(t)} + 0.24 \geq -0.01$ which is imprecise and fails to certify robustness. Note that if $t$ is not the last timestep, the hidden state and the cell state are passed to the next analysis timestep instead of computing the final output.

## 4 AUDIO PROCESSING TRANSFORMERS

As illustrated earlier, the first part of the verification process involves handling of the audio processing stage (performed using MFCC). Here, most matrix multiplication parts can be captured exactly by our abstraction, but MFCC also includes the square operation in the FFT and the logarithm operation in the computation of energy from filterbanks. Thus, to handle these non-linear operations, we need to create new abstract transformers, which we present next. To ensure a minimal loss of precision, our abstract transformers minimize the area between the lower and upper bounds in the input-output plane. This approach of minimizing the area has been used before for other transformers and has been shown to be practically effective in Singh et al. (2018; 2019a).

We denote the set of all variables in the analysis as $\mathcal{X}$. For an element $x \in \mathcal{X}$, we denote the functions corresponding to its linear lower and upper bound as $x^l$ and $x^u$, respectively. These are scalar functions defined on $\mathbb{R}^k$ where $k$ is the number of other variables used to compute $x$. For every element, we also maintain interval bounds, $x \in [l_x, u_x]$. For ease of explanation, we introduce the log transformer followed by the square transformer.

**Log abstract transformer** The logarithm operation is an assignment $y := \log(x)$ where $x, y \in \mathcal{X}$. The output of this operation cannot be captured exactly and we need to carefully introduce an approximation. Here, we first compute the minimum $l_y = \log(l_x)$ and the maximum $u_y = \log(u_x)$, which we use as interval approximation. We define linear lower and upper bound functions $y^l, y^u : \mathbb{R} \rightarrow \mathbb{R}$. Using concavity of the logarithm on any subset of the domain, the lower bound function is a line connecting the points $(l_x, \log(l_x))$ and $(u_x, \log(u_x))$. The upper bound function is chosen as the tangent line to the function minimizing the area between the lower and the upper bound. As a result, we obtain the following bounds as depicted in Fig. 3a:

$$y^l(x) = \log l_x + \frac{x - l_x}{u_x - l_x} \log \frac{u_x}{l_x} \qquad y^u(x) = \frac{2x}{u_x + l_x} - 1 + \log \frac{u_x + l_x}{2}$$

Note that if $l_x \leq 0$, $y^l(x) = -\infty$ since the function is not defined on that domain. We make an exception when $u_x - l_x < 10^{-4}$ to use interval bound for $y^l$ for avoiding the unstable floating point calculation caused by large denominator of $x$ coefficient.

**Square abstract transformer** The square operation is an assignment $y := x^2$ where $x, y \in \mathcal{X}$. Similar to logarithm, this operation is non-linear and cannot be captured exactly in the abstraction. We first compute the interval bounds of the output $l_y$ and $u_y$, and set the minimum value $l_y$ to 0 when $0 \in [l_x, u_x]$ and $\min(l_x^2, u_x^2)$ otherwise. The maximum value $u_y$ is simply $\max(l_x^2, u_x^2)$.

Next, we define linear lower and upper bound functions $y^l, y^u : \mathbb{R} \rightarrow \mathbb{R}$. Using the convexity of the square function, we set the upper bound function $y^u$ to the linear function connecting $(l_x, l_x^2)$ and $(u_x, u_x^2)$. For the lower bound function $y^l$, we have to be delicate: since $x^2 : \mathbb{R} \rightarrow \mathbb{R}^{\geq 0}$, our $y^l$ should also be greater or equal than 0 within any domain. With the principle of minimizing the area between the bounds, we obtain the following bounds for cases as shown in Fig. 3b:

$$y^l(x) = \begin{cases} 4l_x x - 4l_x^2 & 0 \leq l_x < u_x/3 \\ 4u_x x - 4u_x^2 & 0 \geq 3u_x > l_x \\ 0 & l_x \leq 0 \leq u_x \\ (u_x + l_x)x - ((u_x + l_x)/2)^2 & otherwise \end{cases} \qquad y^u(x) = (u_x + l_x)x - u_x l_x$$

## 5 LSTM Transformers

Most prior work on verification of neural networks focuses on feed-forward or convolutional architectures whose operations consist of a sequence of affine transforms and activation functions. However, to perform verification of recurrent architectures, one needs to handle the updates of the recurrent unit. Following the equations updating the LSTM presented in Section 2, we can observe that pre-activations of gates are affine transforms which can be captured exactly in our abstraction. However, the operations updating the cell and hidden states are non-linear and require approximation.

Overall, we have three elementwise products – two are between a sigmoid and a tanh and one between a sigmoid and the identity. A straightforward approach to handling such transforms would be to concretize the polyhedra DeepPoly element to an interval and perform the multiplication using intervals. However, this approach would lose all relations between inputs and incur precision loss in future operations of the analysis. Instead, we design custom binary approximation transformers specifically tailored to handle the elementwise product in the update of the recurrent unit.

**Sigmoid $\odot$ Tanh abstract transformer** We define elementwise multiplication between sigmoid and tanh as an assignment $z := \sigma(x) \tanh(y)$, where $x, y, z \in \mathcal{X}$. As before, our aim is to construct linear bounds $z^l, z^u$. Unlike the previously defined DeepPoly transformers which take as input a single abstract element, this transformer is the first which receives two abstract elements as operands. Hence, the bound functions have the form $z^l, z^u : \mathbb{R}^2 \rightarrow \mathbb{R}$.

Let $z = f(x, y) = \sigma(x) \cdot \tanh(y)$. Our goal is to bound this function between two planes such that the volume between the planes is as small as possible (following our previous heuristic). We first divide the computation into 3 cases based on the signs of $l_y$ and $u_y$. To simplify notation, we introduce the variable assignment in Table 1 so that we can reuse notation across all cases. We first define an *anchor* point $\boldsymbol{a}$: a point in the box $[l_x, u_x] \times [l_y, u_y]$ where function $f$ attains the max/min value defined upon the case. Also we need *reference* point $\boldsymbol{r}$ whereas the plane would meet

| | | $z_x^u$ | $z_y^u$ | $z_x^l$ | $z_y^l$ |
|---|---|---|---|---|---|
| $l_y \geq 0$ | $\Phi$ | min | min | min | min |
| | $\boldsymbol{a}$ | $u_x, u_y$ | $u_x, u_y$ | $l_x, l_y$ | $l_x, l_y$ |
| | $\boldsymbol{r}$ | $l_x, u_y$ | $u_x, l_y$ | $u_x, l_y$ | $l_x, u_y$ |
| $l_y < 0 \leq u_y$ | $\Phi$ | min | min | max | min |
| | $\boldsymbol{a}$ | $u_x, u_y$ | $u_x, u_y$ | $u_x, l_y$ | $u_x, l_y$ |
| | $\boldsymbol{r}$ | $l_x, u_y$ | $l_x, l_y$ | $l_x, l_y$ | $l_x, u_y$ |
| $u_y < 0$ | $\Phi$ | max | min | max | min |
| | $\boldsymbol{a}$ | $l_x, u_y$ | $l_x, u_y$ | $u_x, l_y$ | $u_x, l_y$ |
| | $\boldsymbol{r}$ | $u_x, u_y$ | $l_x, l_y$ | $l_x, l_y$ | $u_y, u_y$ |

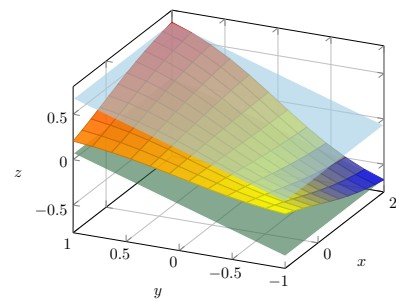

Table 1: Variable reference for the $\sigma(x) \cdot \tanh(y)$ approximation.

Figure 4: Visualization of the bounding planes and $\sigma(x) \cdot \tanh(y)$ curve.

first among the four corners if it is tilted in given direction, other than the anchor. Our transformer computes two candidates planes for the upper bound $(z_x^u, z_y^u)$ and two for the lower bound $(z_x^l, z_y^l)$ as:

$$(z^u | z^l)_x = \Phi\left(D_{\boldsymbol{i}} f(\boldsymbol{a}), \frac{f(\boldsymbol{a}) - f(\boldsymbol{r})}{a_x - r_x}\right)(x - a_x) + f(\boldsymbol{a})$$

$$(z^u | z^l)_y = \Phi\left(D_{\boldsymbol{j}} f(\boldsymbol{a}), \frac{f(\boldsymbol{a}) - f(\boldsymbol{r})}{a_y - r_y}\right)(y - a_y) + f(\boldsymbol{a})$$

where $D_{\boldsymbol{i}} f = \frac{\partial f}{\partial x}$, $D_{\boldsymbol{j}} f = \frac{\partial f}{\partial y}$, and $\Phi$, $\boldsymbol{a}$, $\boldsymbol{r}$ are chosen from Table 1. Finally, we choose bounds from two candidates which minimize the volume between the planes. Fig. 4 is the visualization of this result. We note that Sigmoid ⊙ Identity transformer is handled in the same manner (we can directly apply the same transformer by replacing $f(x, y)$ with $\sigma(x) \cdot y$).

**Theorem 1.** *Our Sigmoid ⊙ Tanh transformer is optimal (it minimizes the volume between the lower and upper plane) under the assumptions: (1) bounding planes are tangent to the curve at the respective anchor points, (2) bounding planes are parallel to either $x$ or $y$ axis. Planes computed using our transformer result in a volume strictly smaller than those computed using interval analysis.*

We show the proof in Appendix B. Our assumptions are based on the following reasoning. The first assumption is need so that our transformer produces bounds strictly better than interval analysis. Unless $z^u$ passes through the point $(u_x, u_y)$, the concrete upper bound would be larger than $\sigma(u_x) \cdot \tanh(u_y)$, making it worse than an interval bound(analogously for the lower bound). The second assumption enables computation of optimal planes in $O(1)$ while without this assumption one needs to solve a non-convex optimization problem which is not feasible at the scale of networks we consider.

## 6 EXPERIMENTS

We now evaluate the effectiveness of DAC on several datasets and neural architectures. All our transformers are implemented in C (for performance) and exposed to the verifier using a Python interface. We will publicly release the datasets, trained networks and source code of DAC. Verification is performed on a Intel(R) Core(TM) i9-9900K CPU @ 3.60GHz using 6 cores for each instance.

**Experimental setup** We evaluate on audio classification benchmarks: Free Spoken Digit Dataset (FSDD) (Jackson, 2019) and Google Speech Commands (GSC) (Warden, 2018). FSDD contains 2,000 samples of spoken digits while GSC is a more complex dataset containing 65,000 utterances of 30 short words spoken by thousands of different people. State-of-the-art accuracy on GSC is 96.9% by de Andrade et al. (2018) using attention RNN, while our verified model achieves 89%.

**Robustness evaluation** Following Singh et al. (2019b), we define success rate of verification as the ratio between certified samples and correctly predicted ones. We randomly shuffled the test data and then, for every experiment, inferred labels one by one until the number of correctly classified samples reached 100. We report the number of provably correct samples out of these 100 as our provability. As a baseline, we consider a verification method based on interval analysis, which only derives neuronwise lower and upper bounds using the previous layer's bounds. We used networks with

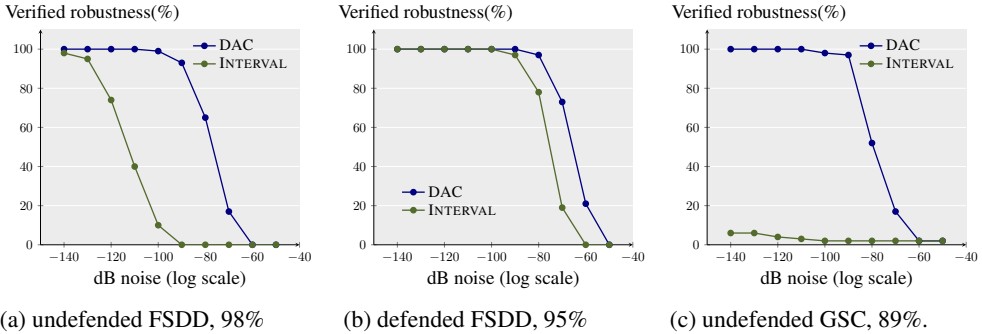

(a) undefended FSDD, 98%     (b) defended FSDD, 95%     (c) undefended GSC, 89%.

Figure 5: Robustness evaluation on FSDD and GSC datasets, using DAC and Interval analysis.

$h = 40$ hidden neurons and $p = 10$ filters for both FSDD and GSC. Our results are shown in Fig. 5. They suggest that DAC substantially outperforms interval analysis due to better approximations of key functions in both audio preprocessing and neural network stages.

**Effect of back-substitution depth** In this experiment, we vary the depth of back-substitution used in DAC and study its effect on our performance. We run this experiment on an FSDD network and report the results in Table 2. We observe that increasing the back-substitution depth increases the provability. This is due to the fact that we benefit from cancellation of common terms in the propagated expressions, as demonstrated in our example in Section 3. However, this comes at a cost of exponential increase in runtime. Thus, we choose to use depth 3 in order to get high verification rate while still having reasonable verification speed.

| Depth | Provability(%) | Run Time(s) |
|---|---|---|
| 0 | 27 | 8.33 |
| 1 | 35 | 18.95 |
| 2 | 37 | 40.83 |
| 3 | 39 | 87.92 |
| 4 | 40 | 166.19 |
| 5 | 41 | 328.92 |

Table 2: Effect of back-substitution depth on the performance.

**Provable defense for audio classifiers** We also for the first time trained audio classifiers to be provably robust against noise-based perturbations. Our training follows Mirman et al. (2018); Gowal et al. (2018) – we perturb the input signal and propagate interval bounds through the audio processing and LSTM stages. To train, we combine standard loss with the worst case loss obtained using interval propagation. The resulting network shown in Fig. 5b achieves 80% of provability for -80 dB even with the imprecise intervals, outperforming the undefended network. Even though this network was specifically trained to be verifiable using intervals, DAC still outperforms intervals and proves significantly more robustness properties. Also note that defended network has lower accuracy of 95%, compared to the 98% accuracy of the baseline.

**Experimental comparison with prior work** POPQORN (Ko et al., 2019) also proposes a method to certify recurrent neural networks by propagating linear bounds. One of the key differences with our work is the approximation of $\sigma(x) \cdot \tanh(y)$ using linear bounds. We found that, in practice, optimization approach used by POPQORN produces approximations of slightly smaller volume than our LSTM transformer (although non-comparable). However, this smaller volume comes at a high cost in runtime. We tried to integrate POPQORN bounds into our verification framework, however we found it not feasible for our audio tasks as it increased our end-to-end runtime $200\times$ on one of the smaller networks. In contrast, DAC takes only 1-2 minutes on average to verify a single example containing multiple frames. Indeed, our observation is consistent with the evaluation of POPQORN in the paper where it takes hours on a GPU for verifying NLP and vision tasks only for a single frame.

## 7 CONCLUSION

We presented the first verifier for certifying audio classifiers. The key idea was to create abstract transformers for non-linear operations used in the audio processing stage and the recurrent network. These transformers compute an optimal (area-wise) approximation under assumptions representable in the underlying convex relaxation and enable sound handling of the entire pipeline. Our evaluation shows that DAC is practically effective and achieves high verification rates on different datasets.

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

## A   Soundness Proof of Sigmoid $\odot$ Tanh Abstract Transformer

Suppose $x$ and $y$ are the abstract variable bounded by $[l_x, u_x]$, $[l_y, u_y]$ respectively. This suggested transformer finds the bounding planes of the function $f(x, y) = \sigma(x) \tanh(y)$. Let $D_i f = \frac{\partial f}{\partial x}$ and $D_j f = \frac{\partial f}{\partial y}$.

1. $l_y \geq 0$
   Choose the upper bound of the function between $z_x^u$ and $z_y^u$, where

   $$z_x^u(x, y) = \min\left( D_i f(u_x, u_y), \frac{f(u_x, u_y) - f(l_x, u_y)}{u_x - l_x} \right)(x - u_x) - f(u_x, u_y)$$

   $$z_y^u(x, y) = \min\left( D_j f(u_x, u_y), \frac{f(u_x, u_y) - f(u_x, l_y)}{u_y - l_y} \right)(y - u_y) - f(u_x, u_y)$$

   by the smaller volume under the each plane.

   **Claim 1:** $z_x^u(x, y) \geq f(x, y)$ in $(x, y) \in [l_x, u_x] \times [l_y, u_y]$.

   Let $x_1 < x_2 \in [l_x, u_x]$ and $y_1 < y_2 \in [l_y, u_y]$. Then for any $x, y$, $f(x, y_1) < f(x, y_2)$ and $f(x_1, y) < f(x_2, y)$. Thus, since $z_x^u$ is independent to $y$, it is sufficient to show $z_x^u(x, \cdot) \geq f(x, u_y)$.

   We can easily know that $f(x, u_y)$ is concave at $x \geq 0$ and convex at $x \leq 0$ by the second derivation of $f$.

   (a) Consider the case of $u_x > 0$. Let $x_0$ be the x coordinate of the crossing of $f(x, u_y)$ and $g(x) = D_i f(u_x, u_y)(x - u_x) - f(u_x, u_y)$. From the convexity of $f(x, u_y)$ in $x < 0$, $g(0) > f(0, u_y)$, and $g'(x) > 0$, there exists a single $x_0 < 0$. If $l_x > x_0$, the slope of $z_x^u(x, \cdot)$ becomes $D_i f(u_x, u_y)$, and $z_x^u$ and $g$ become identical. Hence, $f(x, u_y) \leq g(x) = z_x^u(x, \cdot)$. If $l_x \leq x_0$, the slope of $z_x^u(x, \cdot)$ becomes $\frac{f(u_x, u_y) - f(l_x, u_y)}{u_x - l_x}$. Then $z_x^u$ and $g$ shares a point of $u_x$ but $z_x^u$ has smaller slope, so $g(x) \leq z_x^u(x, \cdot)$ in $[l_x, u_x]$. Also from $f(l_x, u_y) = z_x^u(l_x, \cdot)$ and $f(0, u_y) < z_x^u(0, \cdot)$, $f(x, u_y) \leq z_x^u(x, \cdot)$. Thus, the claim holds when $u_x > 0$.

   (b) Another case of $u_x \leq 0$, x coefficient of $z_x^u$ will always be $\frac{f(u_x, u_y) - f(l_x, u_y)}{u_x - l_x}$. Again, by convexity of $f(x, u_y)$, $z_x^u(x, \cdot) \geq f(x, u_y)$.

   Hence $z_x^u(x, \cdot) \geq f(x, u_y) \geq f(x, y)$ holds in $(x, y) \in [l_x, u_x] \times [l_y, u_y]$. Claim proved. With analogous steps, $z_y^u$ can be shown to cover above the curve. Choosing the plane with smaller volume underneath it allows to minimize the expected difference between the true curve and the upper bound plane under the randomly chosen domain.

   For the lower bound, choose the upper bound of the function between $z_x^l$ and $z_y^l$, where

   $$z_x^l(x, y) = \min\left( D_i f(l_x, l_y), \frac{f(l_x, l_y) - f(u_x, l_y)}{l_x - u_x} \right)(x - l_x) - f(l_x, l_y)$$

   $$z_y^l(x, y) = \min\left( D_j f(l_x, l_y), \frac{f(l_x, l_y) - f(l_x, u_y)}{l_y - u_y} \right)(y - l_y) - f(l_x, l_y)$$

   by the larger volume under the each plane.

   **Claim 2:** $z_x^l(x, y) \leq f(x, y)$ in $(x, y) \in [l_x, u_x] \times [l_y, u_y]$.

   It is sufficient to show $z_x^l(x, \cdot) \geq f(x, l_y)$ since $f(x, y_1) \leq f(x, y_2)$ for $y_1 \leq y_2$.

   (a) Consider the case of $l_x < 0$. Let $x_0$ be the x coordinate of the crossing of $f(x, l_y)$ and $g(x) = D_i f(l_x, l_y)(x - l_x) - f(l_x, l_y)$. From the concavity of $f(x, l_y)$ in $x > 0$, $g(0) < f(0, l_y)$, and $g'(x) > 0$, there exists a single $x_0 > 0$. If $l_x < x_0$, the slope of $z_x^l(x, \cdot)$ becomes $D_i f(l_x, l_y)$, and $z_x^l$ and $g$ become identical. Hence, $f(x, l_y) \leq g(x) = z_x^l(x, \cdot)$. If $l_x \geq x_0$, the slope of $z_x^l(x, \cdot)$ becomes $\frac{f(l_x, l_y) - f(u_x, l_y)}{l_x - u_x}$. Then $z_x^l$ and $g$ shares a point of $l_x$ but $z_x^l$ has smaller slope, so $g(x) \geq z_x^l(x, \cdot)$ in $[l_x, u_x]$. Also from $f(u_x, l_y) = z_x^l(u_x, \cdot)$ and $f(0, l_y) > z_x^l(0, \cdot)$, $f(x, l_y) \geq z_x^l(x, \cdot)$. Thus, the claim holds when $u_x < 0$.

(b) Another case of $l_x \geq 0$, x coefficient of $z_x^l$ will always be $\frac{f(l_x, l_y) - f(u_x, l_y)}{l_x - u_x}$. Again, by convexity of $f(x, l_y)$, $z_x^l(x, \cdot) \leq f(x, l_y)$.

Hence $z_x^l(x, \cdot) \leq f(x, l_y) \geq f(x, y)$ holds in $(x, y) \in [l_x, u_x] \times [l_y, u_y]$. Claim proved.

With analogous steps, $z_y^l$ can be shown to lie under the curve. Choosing the plane with larger volume underneath it allows to minimize the expected difference between the true curve and the lower bound plane under the randomly chosen domain.

2. $l_y \leq 0 \leq u_y$

The proof of upper bounds will follow the same steps with the first case. $z_x^u$ in this case is exactly same as before, but since $f(x, y)$ goes below 0 when $y < 0$, $z_y^u$ has to anchor at $(l_x, l_y)$ instead of $(u_x, l_y)$ since $f(l_x, l_y) \geq f(u_x, l_y)$ and convexity of $f$ in the region. The proof steps do not differ much from the previous proofs.

Again, the proof for lower bound is similar as before, but note that $z_x^l$ needs to choose *maximum* between the two slopes. This is due to the sign of the values. Since $f(u_x, l_y) < 0$ is the minimum in the region and it grows along $x$ gets smaller, both $D_\mathbf{i} f(u_x, l_y)$ and $\frac{f(u_x, l_y) - f(l_x, l_y)}{u_x - l_x}$ are less than zero.

3. $0 \geq u_y$

We will not provide the proof since this case is symmetric to the first case.

## B   PROOF OF THEOREM 1

**Theorem 1.** (copied) *Our Sigmoid $\odot$ Tanh transformer is optimal (it minimizes the volume between the lower and upper plane) under the assumptions: (1) bounding planes are tangent to the curve at the respective anchor points, (2) bounding planes are parallel to either $x$ or $y$ axis. Planes computed using our transformer result in a volume strictly smaller than those computed using interval analysis.*

*Proof.* First note that $z_x^u, z_y^u, z_x^l, z_y^l$ are sound bounds by Appendix A. Assume $z_x^u$ is chosen as the upper bound. W.l.o.g., suppose there exists another sound candidate upper bound $\zeta_x^u$, satisfying (1) and (2), with smaller volume underneath it than $z_x^u$. Then $\zeta_x^u$ would have larger absolute slope on $x$ than $z_x^u$. However, the slope of $z_x^u$ is chosen as stiff as possible while remaining the soundness (Appendix A to see in detail). Hence, $\zeta_x^u$ would not be sound anymore with larger absolute coefficient. Contradict to the existence of sound $\zeta_x^u$. Other cases can be proved with similar logic as well. $\square$

## C   VERIFICATION BY EXAMPLE

We assume the last sound frame $\boldsymbol{s}^{(t)}$ consists of two elements $s_1^{(t)}$ and $s_2^{(t)}$ such that: $s_1^{(t)} \in [0, 1]$ and $s_2^{(t)} \in [-2, -1]$. These constraints capture a noise perturbation of the sound and are depicted in the white box in the left part of Fig. 2. We describe analysis only at the last timestep $t$ (the same process is repeated for every timestep).

We note that the DeepPoly abstraction (which we build on in this work) maintains four constraints for every element: a lower and upper constraints which are linear in terms of the previous elements as well as two interval constraints. This abstraction is exact for affine transformations, however, to handle non-linear operations, one has to create new abstract transformers. In our work we introduce such transformers and formally describe their operation in the next sections. In what follows, we show their effect on our running example. For the presentation below, our figure shows the linear constraints obtained by the verifier, but to avoid visual clutter, we do not show the two interval constraints (however, we do list them in the text).

**Fast Fourier Transform**   The first operation in the audio processing stage is the Fast Fourier Transform (FFT) of the pre-emphasized input signal. It is a two-step process shown in dashed boxes in Fig. 2. The preemphasis is in fact an affine transform so we perform it jointly with the affine transform in the FFT. As the composition of two affine transforms is again affine, this

amounts to a single affine transform on the input. In our example, the composed affine transform is $y_1^{(t)} = s_1^{(t)} - s_2^{(t)}$ and $y_2^{(t)} = 2s_2^{(t)}$. Affine transforms can be captured exactly and here we obtain e.g., $s_1^{(t)} - s_2^{(t)} \le y_1^{(t)} \le s_1^{(t)} - s_2^{(t)}$. We also obtain the interval bounds $y_1^{(t)} \in [1,3]$ and $y_2^{(t)} \in [-4,-2]$.

The next step in the computation of FFT is elementwise square of $\boldsymbol{y}^{(t)}$. We denote this operation as $\theta_1^{(t)} = (y_1^{(t)})^2$ and $\theta_2^{(t)} = (y_2^{(t)})^2$. The square operation is non-linear and cannot be captured exactly which means that we need to decide how to lose precision. Here, we apply our new square abstract transformer (formally defined in Section 4) which provides an optimal linear lower and upper bounds of the square function in terms of area. After applying this transformer, we obtain the bounds:

$$4y_1^{(t)} - 4 \le \theta_1^{(t)} \le 4y_1^{(t)} - 3 \qquad -6y_2^{(t)} - 9 \le \theta_2^{(t)} \le -6y_2^{(t)} - 8$$

The interval bounds are calculated by:

$$\theta_1^{(t)} \ge 4y_1^{(t)} - 4 = 4(s_1^{(t)} - s_2^{(t)}) - 4 \ge 4(0 - (-1)) - 4 = 0$$
$$\theta_1^{(t)} \le 4y_1^{(t)} - 3 = 4(s_1^{(t)} - s_2^{(t)}) - 3 \le 4(1 - (-2)) - 3 = 9$$
$$\theta_2^{(t)} \ge -6y_2^{(t)} - 9 = -6(2s_2^{(t)}) - 9 \ge -6(2 \cdot (-1)) - 9 = 3$$
$$\theta_2^{(t)} \le -6y_2^{(t)} - 8 = -6(2s_2^{(t)}) - 8 \le -6(2 \cdot (-2)) - 8 = 16$$

Hence, $\theta_1^{(t)} \in [0,9]$ and $\theta_2^{(t)} \in [3,16]$. Note those bounds are not exact but sound; even $y_1^{(t)} \in [1,3]$ and $y_2^{(t)} \in [-4,-2]$, DAC calculates the concrete lower and upper bound with the expression to be consistent with other transformers, so the bounds might not be exact as $[1,9]$ and $[4,16]$. In practice, FFT is followed by another affine transform which adds together the real and complex component of each element, but we omit this for clarity as it is captured as before, without loss of precision.

**Filterbanks Transform** Our analysis continues with the computation of filter banks of the input, shown in solid boxes in Fig. 2. The first step is an affine transform and in our example we use: $\tilde{\psi}_1^{(t)} = 2\theta_1^{(t)} + \theta_2^{(t)}$ and $\tilde{\psi}_2^{(t)} = \theta_1^{(t)} + 3\theta_2^{(t)}$. Our abstraction is again exact here and additionally computes interval bounds $\tilde{\psi}_1^{(t)} \in [3,34]$ and $\tilde{\psi}_2^{(t)} \in [9,57]$. The final step in this transform is the elementwise logarithm of the input. We denote the operation as $\psi_1^{(t)} = \log(\tilde{\psi}_1^{(t)})$ and $\psi_2^{(t)} = \log(\tilde{\psi}_2^{(t)})$. As this is again a non-linear operation, we apply our new log transformer so to obtain:

$$0.0783\tilde{\psi}_1^{(t)} + 0.8637 \le \psi_1^{(t)} \le 0.0541\tilde{\psi}_1^{(t)} + 1.9178$$
$$0.0384\tilde{\psi}_2^{(t)} + 1.8511 \le \psi_2^{(t)} \le 0.0303\tilde{\psi}_2^{(t)} + 2.4965$$

The interval bounds are calculated by:

$$\begin{aligned}
\psi_1^{(t)} &\ge 0.0783\tilde{\psi}_1^{(t)} + 0.8637 = 0.0783(2\theta_1^{(t)} + \theta_2^{(t)}) + 0.8637 \\
&\ge 0.0783(2(4y_1^{(t)} - 4) + (-6y_2^{(t)} - 9)) + 0.8637 \\
&= 0.0783(2(4(s_1^{(t)} - s_2^{(t)}) - 4) + (-6(2s_2^{(t)}) - 9) + 0.8637 \\
&= 0.6264 s_1^{(t)} - 1.566 s_2^{(t)} - 0.4674 \ge 1.0986 \\
\psi_1^{(t)} &\le 0.0541\tilde{\psi}_1^{(t)} + 1.9178 = 0.0541(2\theta_1^{(t)} + \theta_2^{(t)}) + 1.9178 \\
&\le 0.0541(2(4y_1^{(t)} - 3) + (-6y_2^{(t)} - 8)) + 1.9178 \\
&= 0.0541(2(4(s_1^{(t)} - s_2^{(t)}) - 3) + (-6(2s_2^{(t)}) - 8)) + 1.9178 \\
&= 0.4328 s_1^{(t)} - 1.082 s_2^{(t)} + 1.1604 \le 3.7572
\end{aligned}$$

$$\psi_2^{(t)} \geq 0.0384\tilde{\psi}_2^{(t)} + 1.8511 = 0.0384(\theta_1^{(t)} + 3\theta_2^{(t)}) + 1.8511$$
$$\geq 0.0384((4y_1^{(t)} - 4) + 3(-6y_2^{(t)} - 9)) + 1.8511$$
$$= 0.0384((4(s_1^{(t)} - s_2^{(t)}) - 4) + 3(-6(2s_2^{(t)}) - 9)) + 1.8511$$
$$= 0.1536s_1^{(t)} - 1.536s_2^{(t)} + 0.6607 \geq 2.1967$$
$$\psi_2^{(t)} \leq 0.0303\tilde{\psi}_2^{(t)} + 2.4965 = 0.0303(\theta_1^{(t)} + 3\theta_2^{(t)}) + 2.4965$$
$$\leq 0.0303((4y_1^{(t)} - 3) + 3(-6y_2^{(t)} - 8)) + 2.4965$$
$$= 0.0303((4(s_1^{(t)} - s_2^{(t)}) - 3) + 3(-6(2s_2^{(t)}) - 8)) + 2.4965$$
$$= 0.1212s_1^{(t)} - 1.212s_2^{(t)} + 1.6784 \leq 4.2236$$

In other words, $\psi_1^{(t)} \in [1.0986, 3.7572]$ and $\psi_2^{(t)} \in [2.1967, 4.2236]$. Again, these bounds are sound since $[\log 3, \log 34] \subset [1.0986, 3.7572]$, $[\log 9, \log 57] \subset [2.1967, 4.2236]$.

**Discrete Cosine Transform and ReLU**  After the Filterbanks Transform, the input is passed through the Discrete Cosine Transform (DCT) followed by a Lifting operation. The analysis result of these steps is then provided as an input to a fully connected (FC) layer followed by a ReLU activation. To ease presentation, in our example we combine DCT, Lifting and FC layer in a single affine transform: $\tilde{x}_1^{(t)} = \psi_1^{(t)} - \psi_2^{(t)} + 4$ and $\tilde{x}_2^{(t)} = 4\psi_1^{(t)} - \psi_2^{(t)}$. We show this transform in a dotted box in Fig. 2. This is again captured exactly in our abstraction along with interval bounds $\tilde{x}_1^{(t)} \in [0.875, 5.5605]$, $\tilde{x}_2^{(t)} \in [0.1708, 12.8321]$. The affine transform is followed by a ReLU activation $x_1^{(t)} = \max(0, \tilde{x}_1^{(t)})$, $x_2^{(t)} = \max(0, \tilde{x}_2^{(t)})$. In general, we use the ReLU transformer defined in Singh et al. (2019a), which for this example produces: $x_1^{(t)} = \tilde{x}_1^{(t)}$ and $x_2^{(t)} = \tilde{x}_2^{(t)}$.

**LSTM analysis**  After the verifier completes the audio processing stage, its output is passed as input to the LSTM cell. This cell also receives a hidden state and a cell state from the previous timestep (shown as a blue box in Fig. 2). In our example, we assume these are given as: $h_1^{(t-1)}, h_2^{(t-1)}, c_1^{(t-1)}, c_2^{(t-1)} \in [0.9, 1]$. We note that these elements usually have different interval bounds, but for simplicity in our example we use the same intervals. We focus on the first neuron in the LSTM and compute pre-activations in our example for the forget gate $f_1^{(t)} = x_1^{(t)}/4$, input gate $i_1^{(t)} = 2h_2^{(t-1)}$, output gate $o_1^{(t)} = h_1^{(t-1)}$ and cell state $\tilde{c}^{(t)} = h_1^{(t-1)} - x_2^{(t)}/4$.

In order to update the cell state, we need to compute: $c_1^{(t)} = \sigma(f_1^{(t)}) \cdot c_1^{(t-1)} + \sigma(i_1^{(t)}) \cdot \tanh(\tilde{c}_1^{(t)})$. The left and right summands are computed using our new binary abstract transformers for $\sigma(x) \cdot y$ and $\sigma(x) \cdot \tanh(y)$ expressions, respectively. Applying our abstract transformers on the summands produces the following bounds:

$$0.1891f_1^{(t)} + 0.4576 \leq \sigma(f_1^{(t)}) \cdot c_1^{(t-1)} \leq 0.1596f_1^{(t)} + 0.5787$$
$$0.0341\tilde{c}_1^{(t)} - 0.7847 \leq \sigma(i_1^{(t)}) \cdot \tanh(\tilde{c}_1^{(t)}) \leq 0.3945\tilde{c}_1^{(t)} + 0.2769$$

Summing up the above inequalities grants the upper and lower bounds of $c_1^{(t)}$ in terms of $\boldsymbol{x}^{(t)}$, $\boldsymbol{h}^{(t-1)}$, and $\boldsymbol{c}^{(t-1)}$:

$$c_1^{(t)} \geq 0.1891f_1^{(t)} + 0.0341\tilde{c}_1^{(t)} - 0.3271$$
$$= 0.1891(x_1^{(t)}/4) + 0.0341(h_1^{(t-1)} - x_2^{(t)}/4) - 0.3271$$
$$\geq -0.3644$$
$$c_1^{(t)} \leq 0.1596f_1^{(t)} + 0.3945\tilde{c}_1^{(t)} + 0.8556$$
$$= 0.1596(x_1^{(t)}/4) + 0.3945(h_1^{(t-1)} - x_2^{(t)}/4) + 0.8556$$
$$\leq 1.4551$$

and get $c_1^{(t)} \in [-0.3644, 1.4551]$. We omit the back-substitution steps since we do not lose precision over the affine transformations from $\boldsymbol{x}^{(t)}$, and note that this calculation is based on the assumption

that the bounds of $\boldsymbol{h}^{(t)}$ and $\boldsymbol{c}^{(t-1)}$ are given constant. In practice, we recursively apply the process for those vectors to express the bounds in terms of $\boldsymbol{x}^{(t)}, \boldsymbol{x}^{(t-1)}, \cdots$.

The next hidden state is computed as $h_1^{(t)} = \sigma(o_1^{(t)}) \cdot \tanh(c_1^{(t)})$. We apply our abstract transformer for the $\sigma(x) \cdot \tanh(y)$ expression, we obtain:

$$
\begin{aligned}
h_1^{(t)} &\geq 0.4906 c_1^{(t)} - 0.0764 \\
&\geq 0.4906(0.1891 f_1^{(t)} + 0.0341 \tilde{c}_1^{(t)} - 0.3271) - 0.0764 \\
&\geq -0.2551 \\
h_1^{(t)} &\leq 0.1433 c_1^{(t)} + 0.4471 \\
&\leq 0.1433(0.1596 f_1^{(t)} + 0.3945 \tilde{c}_1^{(t)} + 0.8556) + 0.4471 \\
&\leq 0.6556
\end{aligned}
$$

i.e., $h_1^{(t)} \in [-0.2551, 0.6556]$. An analogous computation is performed for $h_2^{(t)}$. In this example we will assume $h_2^{(t)} = 0$ concretely.

**Robustness certification using DeepPoly**  The hidden states in the LSTM at the final timestep are passed through a fully connected layer without activation. In our example we assume that logits after the layer are obtained with the following formula: $l_1 = h_1^{(t)} + 0.24$ and $l_2 = h_2^{(t)}$. These are shown in the final box of Fig. 2. To certify that the neural network classifies the input to class 1, we need to prove that $l_1 - l_2 > 0$. We now apply the same back-substitution technique as Singh et al. (2019a):

$$
\begin{aligned}
l_1 - l_2 &= h_1^{(t)} - h_2^{(t)} + 0.24 \\
&\geq 0.4906(0.1891 f_1^{(t)} + 0.0341 \tilde{c}_1^{(t)} - 0.3271) - 0.0764 + 0.24 \\
&= 0.0927 f_1^{(f)} + 0.0167 \tilde{c}_1^{(t)} + 0.0031 \\
&= 0.0927(x_1^{(t)}/4) + 0.0167(h_1^{(t-1)} - x_2^{(t)}/4) + 0.0031 \\
\text{(applying concrete bound to } h_1^{(t-1)} \text{ for demonstration)} \quad &\geq 0.0231 x_1^{(t)} - 0.0042 x_2^{(t)} + 0.0181 \\
&= 0.0231(\psi_1^{(t)} - \psi_2^{(t)} + 4) - 0.0042(4\psi_1^{(t)} - \psi_2^{(t)}) + 0.0181 \\
&= 0.0063 \psi_1^{(t)} - 0.0189 \psi_2^{(t)} + 0.1105 \\
\text{halt the computation and plug in the bounds} \quad &\geq 0.0063 \cdot 1.0986 - 0.0189 \cdot 4.2236 + 0.1105 \\
&\geq 0.0375
\end{aligned}
$$

For the purpose of demonstration, we stop here and plug in previously obtained interval bounds for $\boldsymbol{\psi}^{(t)}$. As this value is greater than 0, robustness is established.

## D  MATHEMATICAL DERIVATION OF LSTM TRANSFORMER

LSTM cell receives not only the resulting neurons from the pre-processing stage but a hidden state and a cell state from the previous timestep. To incorporate with all those incoming values, we extend the scope of the LSTM equations defined in Section 2. Let $d$ be the dimension of the input and $h$ be the dimension of states of the cell. To standardize the operations performed within the LSTM cell, we append $\mathbf{0}_{d \times d} = [0]_{d \times d}$ and $\mathbf{0}_d = [0]_d$ below to each kernel and bias respectively to express the formula as

$$
\begin{aligned}
\boldsymbol{f}_0^{(t)} &= [\boldsymbol{x}^{(t)}, \boldsymbol{h}^{(t-1)}, \boldsymbol{c}^{(t-1)}]\boldsymbol{W}_f' + \boldsymbol{b}_f' & \boldsymbol{o}_0^{(t)} &= [\boldsymbol{x}^{(t)}, \boldsymbol{h}^{(t-1)}, \boldsymbol{c}^{(t-1)}]\boldsymbol{W}_o' + \boldsymbol{b}_o' \\
\boldsymbol{i}_0^{(t)} &= [\boldsymbol{x}^{(t)}, \boldsymbol{h}^{(t-1)}, \boldsymbol{c}^{(t-1)}]\boldsymbol{W}_i' + \boldsymbol{b}_i' & \tilde{\boldsymbol{c}}_0^{(t)} &= [\boldsymbol{x}^{(t)}, \boldsymbol{h}^{(t-1)}, \boldsymbol{c}^{(t-1)}]\boldsymbol{W}_{\tilde{c}}' + \boldsymbol{b}_{\tilde{c}}'
\end{aligned}
\tag{1}
$$

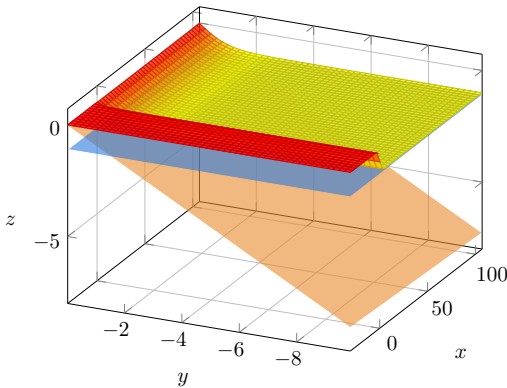

Figure 6: $\sigma(x) \cdot \tanh(y)$ boundings on $[-30.7, 107] \times [-9.87, -0.02]$. Orange plane is obtained from POPQORN and blue plane is from DAC.

Then the cell state and hidden state computations can be rewritten as

$$\boldsymbol{c}^{(t)} = \sigma(\boldsymbol{f}_0^{(t)}) \odot \boldsymbol{c}^{(t-1)} + \sigma(\boldsymbol{i}_0^{(t)}) \odot \tanh(\tilde{\boldsymbol{c}}_0^{(t)})$$
$$= \sigma([\boldsymbol{x}^{(t)}, \boldsymbol{h}^{(t-1)}, \boldsymbol{c}^{(t-1)}]\boldsymbol{W}_f' + \boldsymbol{b}_f') \odot ([\boldsymbol{x}^{(t)}, \boldsymbol{h}^{(t-1)}, \boldsymbol{c}^{(t-1)}][\boldsymbol{0}_{h \times (d+h)}, \boldsymbol{I}_h]^T)$$
$$+ \sigma([\boldsymbol{x}^{(t)}, \boldsymbol{h}^{(t-1)}, \boldsymbol{c}^{(t-1)}]\boldsymbol{W}_i' + \boldsymbol{b}_i') \odot \tanh([\boldsymbol{x}^{(t)}, \boldsymbol{h}^{(t-1)}, \boldsymbol{c}^{(t-1)}]\boldsymbol{W}_{\tilde{c}}' + \boldsymbol{b}_{\tilde{c}}') \quad (2)$$
$$\boldsymbol{h}^{(t)} = \sigma(\boldsymbol{o}_0^{(t)}) \odot \tanh(\boldsymbol{c}^{(t)})$$
$$= \sigma([\boldsymbol{x}^{(t)}, \boldsymbol{h}^{(t-1)}, \boldsymbol{c}^{(t-1)}]\boldsymbol{W}_o' + \boldsymbol{b}_o') \odot \tanh(\boldsymbol{c}^{(t)}) \quad (3)$$

As is described in detail in Section 5, $f(x, y) = \sigma(x) \cdot \tanh(y)$ and $g(x, y) = \sigma(x) \cdot y$ can be bounded with two planes regarding the operands' lower and upper bounds. In other words, for any given two real values, $x, y \in \mathbb{R}$, $x \in [l_x, u_x]$, $y \in [l_y, u_y]$, our finding guarantees the planar bounds of $f$ and $g$ with the coefficients built by linear combination of $l_x, u_x, l_y, u_y$. For a single neuron $c_j^{(t)}$ in $\boldsymbol{c}^{(t)}$, the upper bound can be expressed from Eq. (2):

$$c_j^{(t)} = \sigma([\boldsymbol{x}^{(t)}, \boldsymbol{h}^{(t-1)}, \boldsymbol{c}^{(t-1)}][\boldsymbol{W}_f']_{:,j} + [\boldsymbol{b}_f']_j) \cdot ([\boldsymbol{x}^{(t)}, \boldsymbol{h}^{(t-1)}, \boldsymbol{c}^{(t-1)}][\boldsymbol{0}_{d+h}, [\boldsymbol{I}_h]_{j,:}]^T)$$
$$+ \sigma([\boldsymbol{x}^{(t)}, \boldsymbol{h}^{(t-1)}, \boldsymbol{c}^{(t-1)}][\boldsymbol{W}_i']_{:,j} + [\boldsymbol{b}_i']_j) \cdot \tanh([\boldsymbol{x}^{(t)}, \boldsymbol{h}^{(t-1)}, \boldsymbol{c}^{(t-1)}][\boldsymbol{W}_{\tilde{c}}']_{:,j} + [\boldsymbol{b}_{\tilde{c}}']_j)$$
$$\leq A_u([\boldsymbol{x}^{(t)}, \boldsymbol{h}^{(t-1)}, \boldsymbol{c}^{(t-1)}][\boldsymbol{W}_f']_{:,j} + [\boldsymbol{b}_f']_j) + B_u([\boldsymbol{x}^{(t)}, \boldsymbol{h}^{(t-1)}, \boldsymbol{c}^{(t-1)}][\boldsymbol{0}_{d+h}, [\boldsymbol{I}_h]_{j,:}]^T) + C_u$$
$$+ D_u([\boldsymbol{x}^{(t)}, \boldsymbol{h}^{(t-1)}, \boldsymbol{c}^{(t-1)}][\boldsymbol{W}_i']_{:,j} + [\boldsymbol{b}_i']_j) + E_u([\boldsymbol{x}^{(t)}, \boldsymbol{h}^{(t-1)}, \boldsymbol{c}^{(t-1)}][\boldsymbol{W}_{\tilde{c}}']_{:,j} + [\boldsymbol{b}_{\tilde{c}}']_j) + F_u$$
$$= \boldsymbol{x}^{(t)}\boldsymbol{w}_x + \boldsymbol{h}^{(t-1)}\boldsymbol{w}_h + \boldsymbol{c}^{(t-1)}\boldsymbol{w}_c + b \quad (4)$$

where $A_u, \cdots, F_u$ are calculated real values by Section 5, and $\boldsymbol{w}_.$ and $b$ are the simplified linear coefficients of each vector and bias term, respectively. We also can get the upper bound of $h_j^{(t)}$ by plugging Eq. (4) in Eq. (3), and similarly for the lower bounds of those states.

Note these linear bounds makes possible to recursively apply this step to get desired bounds, and can be branched in any depth for the trade-off between the precision and the complexity.

# E  COMPARISON BETWEEN DAC AND POPQORN

In this subsection we provide in-depth comparison between our approach and POPQORN (Ko et al., 2019) which provides another way to bound $\sigma(x) \cdot \tanh(y)$ and $\sigma(x) \cdot y$ non-linearities. Before the experiments, we discuss comparability between three methods to bound the non-linearities: interval, DAC and POPQORN. Then, we perform three different experiments. First, we compare volume of bounds produced by both POPQORN and DAC on a set of synthetic test cases. As these test cases do not reflect the actual test cases which occur during the verification of audio benchmarks, in the second set of experiments we compare the methods on a set of test cases from one of our benchmarks. Finally, in the third experiment we plug in POPQORN bounds into our framework and show its certification performance on the audio benchmarks.

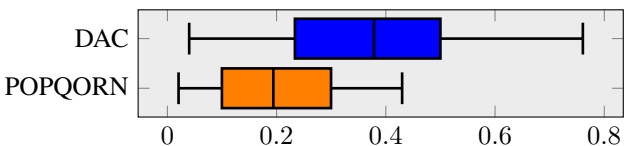

Figure 7: Volume comparison with interval bounds with synthesized data. Excluded outliers.

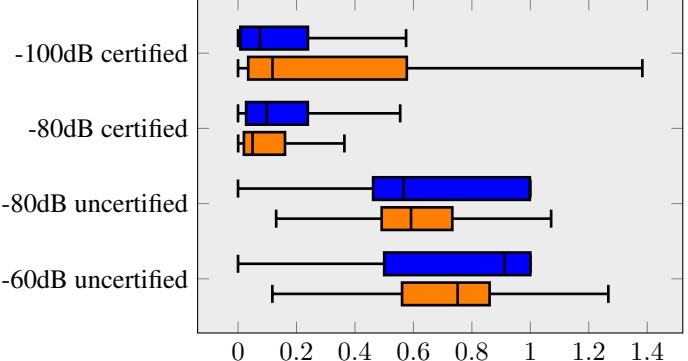

Figure 8: Volume comparison with interval bounds with data from the working pipeline. Excluded outliers.

**Comparability of different bounding methods** We say that methods A and B are pairwise *non-comparable* if there exists an input for which method A produces tighter bound than method B, and vice versa. Given this definition, POPQORN is non-comparable with our method. To demonstrate this, in Fig. 6 we show a case where this behavior is manifested. Here, for $y \leq -1$ POPQORN (shown as orange plane) produces tighter bound than DAC (shown as blue plane). However, for the other entire range of inputs where $-10 \leq y \leq -2$, POPQORN bounds are substantially worse than our bounds. Further, those bounds are even worse than interval bounds and the overapproximation error is not bounded. Contrary to POPQORN, our bounds are always *strictly better* than interval bounds (this is proven in Theorem 1) and distance between the function and our planes is bounded.

**Comparison on synthetic test cases** In this experiment, we compare bounds produced by POPQORN and DAC on a set of synthetic test cases, unrelated to the certification of audio classifiers. Both methods take $l_x, u_x$ and $l_y, u_y$ as bounds for the input to the sigmoid and tanh, respectively. We sample those inputs uniformly from $[-2, 2] \times [-2, 2]$ and compared the volume between the curve $\sigma(x) \cdot \tanh(y)$ and bounding planes produced by both DAC and POPQORN. The volume was computed using 1000 Monde Carlo samples in the set $[l_x, u_x] \times [l_y, u_y]$. Since there are two $\sigma(x) \cdot \tanh(y)$ and one $\sigma(x) \cdot y$ calculation appearing in a single LSTM cell, we run the experiment with such portion of data. In other words, in 67% of experiments we bound $\sigma(x) \cdot \tanh(y)$, and in 33% we bound $\sigma(x) \cdot y$. We sampled 100 such inputs and compared the volumes obtained by POPQORN and DAC with the volume obtained using interval.

The distribution of volumes is shown in Fig. 7. Here, 1 stands for the same volume as interval bounds and values less than 1 indicate performance better than intervals. We conclude that, for this experiment, POPQORN bounds produces smaller volume than our method - 0.2 compared to 0.37. In terms of runtime, POPQORN takes 14.37 seconds on average while the bound calculation of DAC finishes in few milliseconds.

**Direct comparison on test cases from audio benchmarks** The previous experiment may not reflect the actual performance on audio benchmarks that we consider in this work. We uniformly sampled the arguments $l_x, u_x, l_y, u_y$ that our transformer for $\sigma(x) \cdot \tanh(y)$ and $\sigma(x) \cdot y$ was invoked with. We distinguish between test cases corresponding to different perturbation ranges and then further split it into certified and uncertified samples. For each of the cases, Fig. 8 shows the box plot of the distributions of volumes produced by both POPQORN and DAC.

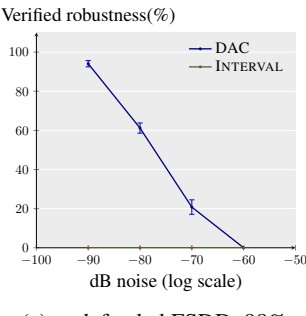 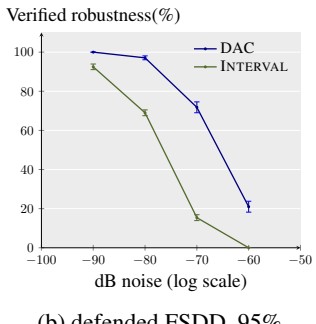

(a) undefended FSDD, 98%          (b) defended FSDD, 95%

Figure 9: Robustness evaluation on FSDD with error bars.

We found that, while overall POPQORN bounds work well in practice, they frequently produce bounds less precise than interval bounds. The reason of this malfunctioning comes from the limitation of gradient descent approach employed by POPQORN. The gradient of the function, which POPQORN uses to search for the optimal bounds, is large when the inputs are distributed near 0. However, if the inputs are far from the origin or are too close to each other, the function curve becomes almost flat, gradients are close to zero and gradient descent has problems converging. Also, in other cases, gradient descent is not guaranteed to find the bounds with minimum volume. Fig. 6 shows one of the examples where POPQORN fails to find lower planar bound which produces minimum volume. On the contrary, the resulting value of $\sigma(x) \cdot \tanh(y)$ is within $[-1, 1]$ regardless of the magnitude of arguments which means that the error produced by intervals is bounded. As our planar bounds are strictly better than intervals, regardless of input conditions, our error is also bounded.

**Plugging in POPQORN bounds into DAC**   We also experimented with using POPQORN bounds instead of our bounds in the pipeline and compared the final adversarial regions with those resulting from DAC. As POPQORN is relatively slow (108 minutes per sample), we performed this experiment only on the first 10 inputs with $-80dB$ perturbation. Using their bounds results in 0 verified samples, while DAC verifies 4 samples. We believe the reason here is the existence of many pathological cases as described in the previous point where gradient descent used by POPQORN converges to suboptimal solution which ends up being worse than interval bounds. These errors are further propagated through each frame and resulting output can not be certified.

## F    SENSITIVITY OF PROVABILITY METRIC

In our experiments, we followed the convention of provability measurement from Singh et al. (2019b). Here, we also provide the result with error bars from 10 independent repetitions with randomly permuted test set. For each repetition, we randomly permute the test set with the different seed and collect the first 100 samples with the correct prediction under zero perturbation from the ordered set. We then count the number of certified inputs from those samples to represent the provability under the given constant $\epsilon$. Fig. 9 shows that provabilities do not differ much from the reported results for multiple experiments.

## G    TRAINING WITH PROVABLE DEFENSE WITH IBP

Here we give more details on our training procedure for provably defended network which follows Mirman et al. (2018); Gowal et al. (2018). Let $\boldsymbol{z}^{LB}(\epsilon)$ and $\boldsymbol{z}^{UB}(\epsilon)$ be the resulting lower and upper bounds for the final logit $\boldsymbol{z} \in \mathbb{R}^d$ under the perturbation size $\epsilon$, with the true label $j \in \{0, \cdots, d-1\}$. We define the *worst-case logits* $\hat{\boldsymbol{z}}(\epsilon)$ as

$$\hat{\boldsymbol{z}}_i(\epsilon) = \begin{cases} \boldsymbol{z}_i^{LB}(\epsilon) & i = j \\ \boldsymbol{z}_i^{UB}(\epsilon) & i \neq j \end{cases}$$

which corresponds to the worst concrete logits under the given input with the predefined perturbation amount. Recall that we say *the input is certified* when this worst-case logits satisfy $j = \arg\max_i \hat{\boldsymbol{z}}_i$.

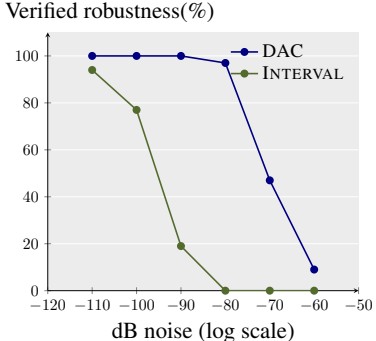

Figure 10: defended GSC, 82%.

| params | FSDD | GSC |
|---|---|---|
| Frame size $N$ | 256 | 512 |
| Frame step $s$ | 200 | 400 |
| Number of filters $p$ | 10 | 10 |
| Hidden neurons $h$ | 40 | 40 |
| Learning rate | 0.001 | 0.001 |
| Batch size | 5 | 50 |

Table 3: Model parameters used in the experiments.

Training loss $L$ is a linear combination of the standard cross-entropy loss $l(\boldsymbol{z}, \boldsymbol{e}^{(j)})$ and the worst-case cross-entropy loss $l(\hat{\boldsymbol{z}}, \boldsymbol{e}^{(j)})$ where $\boldsymbol{e}^{(j)}$ is target one-hot vector, i.e.,

$$L(t') = \kappa(t')l(\boldsymbol{z}, \boldsymbol{e}^{(j)}) + (1 - \kappa(t'))l(\hat{\boldsymbol{z}}(\epsilon(t')), \boldsymbol{e}^{(j)})$$

Note that we set up the $\kappa$ and $\epsilon$ as the function of $t'$. As in Gowal et al. (2018), gradual increase/decrease of these parameters during the training was essential to get desired performance. We set these functions with respect to the number of training epochs $E$ as

$$\kappa(t) = \begin{cases} 1 - t/E & t < E/2 \\ 1/2 & otherwise \end{cases}$$

$$\epsilon(t) = -70 - (200 - 70)^{1 - t/E}.$$

Also, to track the training speed, we increase one by one $t'$ only if we are achieving 85% of standard accuracy and 80% provability at the current setting. The models were built under the $E = 60$.

We also attach the missing result of defended GSC with 82% concrete accuracy in Fig. 10.

## H  MODEL PARAMETERS USED IN THE EXPERIMENTS

We perform the experiments with the same architecture with different model parameters for FSDD and GSC. Table 3 shows the parameters of the models we use. Both defended and undefended networks share the same parameters for each dataset.

