# OpenReview forum: "Certifying Neural Network Audio Classifiers"
_ICLR.cc/2020/Conference — Reject_

### Official Review · AnonReviewer1 · 2019-10-22
**Official Blind Review #1**

**Rating:** 1

**Review:**

This paper proposes an approach for the certification of speech classification neural networks against adversarial perturbations. The network is based on a simple pipeline starting from MFCC to make an utterance level classifier via a last hidden state of an LSTM acoustic model. This approach can perform analysis through this pipeline. I feel that this paper is very difficult to follow because of the lack of background technique explanations based on neural network certification, and the lack of technical surveys of speech recognition and related areas. This paper requires such major restructuring and more surveys to make it in good shape.

Comments
- The authors only list CTC related techniques as state-of-the-art ASR, but state-of-the-art ASR is still based on the HMM/DNN hybrid system or attention-based encoder-decoder/RNN transducer. They seriously lack the surveys o this area. Also several technical terminologies are not common in the speech recognition are (e.g., automated speech recognition --> automatic speech recognition)
- "Additionally, audio systems typically use recurrent architectures (Chiu et al., 2017)": There are a lot of state-of-the-art ASR systems including TDNN (Kaldi), CNN, and transformer. Again the paper does not have enough surveys.
- The font size of the characters in figure 1 is too small.
- I cannot understand why the paper uses MFCC. The community was already moved from MFCC to log mel filterbank. We don't need final DCT.
- Section 2. Threat model: what kind of noises are using?
- Page 3, power operation: Either side must be a conjugate to get the power spectrum.





**Experience Assessment:**

I have read many papers in this area.

**Review Assessment: Checking Correctness Of Derivations And Theory:**

I did not assess the derivations or theory.

**Review Assessment: Checking Correctness Of Experiments:**

I did not assess the experiments.

**Review Assessment: Thoroughness In Paper Reading:**

I read the paper at least twice and used my best judgement in assessing the paper.

---

> ### Author Response · Authors · 2019-11-12
> **Reply for Reviewer#1**
>
> Thank you for the feedback and interest in our work. Below we answer the concerns:
>
>
> Q1: Can you provide background on neural network certification?
>
> A1: Certification of neural networks is an emerging area which aims to prove the robustness of neural networks against adversarial perturbations. For an introduction to the area, we refer the reviewer to related work listed in the introduction, particularly [1] which provides a unifying framework for many of the existing approaches. You can also check the answer we provide to Reviewer 2 for more technical details.
>
> Certification of neural networks is still at an early stage and none of the existing approaches scales to state-of-the-art networks and datasets (e.g. ImageNet). While here we do not directly consider the state-of-the-art architectures you listed, our method for certifying recurrent networks is general and can be applied to those architectures as well. However, certification methods for computer vision tasks have scaled from verifying small networks with 200 neurons to large residual networks with thousands of neurons and we expect the same progress in the audio domain in the future.
>
> Our work is the first one to certify audio classifiers. None of the existing approaches can handle this task and are mostly limited to computer vision. The only alternative approach which can handle recurrent networks, POPQORN, does not scale to the audio benchmarks as we demonstrated in our experiments in Appendix E.
>
>
> Q2: Section 2. Threat model: what kind of noises are using?
>
> A2: Please see the definition of our threat model at the beginning of Section 2.
>
>
> [1] Salman, Hadi, et al. "A convex relaxation barrier to tight robust verification of neural networks.", NeurIPS 2019

---

### Official Review · AnonReviewer2 · 2019-10-23
**Official Blind Review #2**

**Rating:** 6

**Review:**

This paper presents an end-to-end neural network verifier that is specially designed for audio signal processing to certify the robustness of a system when facing noise perturbation.  The approach is based on abstract transformers to deal with non-linearity in the audio signal processing pipeline and LSTM acoustic model. The authors implement the approach in a so-called "deep audio certifier" system and conduct experiments on various datasets and network architectures.  The results seem to be supportive.   The idea is good and the mathematical derivation is meticulous (although appears to be a bit tedious).  This is an interesting paper globally but I have some concerns.

1.  There should be a more thorough introduction on how to verify the robustness of a neural network classifier for noise perturbation.  There should be some background summary such as what are the existing approaches and how to measure the robustness of a neural network classifier, etc..

2. The so-called audio processing pipeline here is actually speech processing pipeline. I am not sure if "audio" is the right term in a strict sense.

3.  This paper is closely related to the POPQORN paper. So it is good to see some in-depth discussion and comparison between the two.  One thing that is not clear to me is that the authors claim POPQORN is very time-consuming but DAC is much faster.  I wonder if the authors can elaborate a bit more on this issue.   What exactly makes POPQORN time-consuming in this case?

P.S. rebuttal read.  I will stay with my score.

**Experience Assessment:**

I have read many papers in this area.

**Review Assessment: Checking Correctness Of Derivations And Theory:**

I assessed the sensibility of the derivations and theory.

**Review Assessment: Checking Correctness Of Experiments:**

I assessed the sensibility of the experiments.

**Review Assessment: Thoroughness In Paper Reading:**

I read the paper at least twice and used my best judgement in assessing the paper.

---

> ### Author Response · Authors · 2019-11-12
> **Reply for Reviewer#2**
>
> Thank you for the feedback and interest in our work. Below we answer the concerns:
>
>
> Q1: There should be a more thorough introduction on how to verify the robustness of a neural network classifier for noise perturbation.  There should be some background summary such as what are the existing approaches and how to measure the robustness of a neural network classifier, etc..
>
> A1: Thank you for the suggestion, we will extend our background section with the description of existing approaches for certification of neural networks. In the meantime, here we provide short summary:
>
> Most existing state-of-the-art scalable certification methods aim to capture all possible behaviors of a neural network using convex relaxations. In this perspective, we treat the input with the predefined perturbation range as a multi-dimensional polyhedron and pass it through the operations within the network.
> The most basic approach here uses interval propagation - it maintains the minimum and maximum possible value for each neuron in the network. More recent work (listed in Section 2) presents more elaborate relaxations to capture the propagation of the initial perturbation through the network.
> Robustness certification is performed by checking whether the neuron corresponding to the true label is always greater than neurons corresponding to other labels with respect to the input region. If this is true, then we can establish the network correctly classifies the input under all possible realized perturbations. We will further elaborate on this in the next revision.
>
>
> Q2: The so-called audio processing pipeline here is actually speech processing pipeline. I am not sure if "audio" is the right term in a strict sense.
>
> A2: Thank you for pointing this out, we will modify our terminology to more precisely match what we describe in the paper.
>
>
> Q3: Can you provide in-depth discussion and comparison between your approach and POPQORN?
>
> A3: Please see our answer to the main points above where we provide answers to these questions and an in-depth comparison between POPQORN and our approach.

---

### Official Review · AnonReviewer3 · 2019-10-27
**Official Blind Review #3**

**Rating:** 3

**Review:**

In this work, the authors study the task of building neural network classifiers for audio tasks which can be certified as being resistant to an adversarial attack. One of the contributions of this work is the development of abstract transformers which can be used for the data processing frontend used in typical audio applications. The work also proposes an abstract transformers for LSTMs which is stated to be much faster to use in practice than previous work.

Overall, this work is interesting and I think it would be a great addition to the conference. The paper is generally well written in the initial sections, and the main ideas are very clearly presented. However, there are a number of missing details, particularly in the final sections which discuss the experimental validation. In its present form, I am rating this work as “weak reject”, but I would increase my scores if the authors can improve the final sections in the revised draft.

Main Comments:
1. While I found section 3 to be useful to get an intuition of the proposed method, I still feel that it could be condensed a bit to add in additional details. For example, the authors don’t describe “back-substitution” in the work, which I believe should be described in the main text.
2. A clarification question: When computing provability, the authors state that “We randomly shuffled the test data and then, for every experiment, inferred labels one by one until the number of correctly classified samples reached 100. We report the number of provably correct samples out of these 100 as our provability.” How sensitive was the provability metric to the choice of these 100 test examples? Was the metric computed by repeatedly sampling 100 test cases, for example?
3. The section on “Provable defense for audio classifiers” was not very clear to me. The authors state that “To train, we combine standard loss with the worst case loss obtained using interval propagation.” I was not clear on what the modified loss is. Could the authors please clarify this in the text, preferably a mathematical formulation? Also, I’m curious why these experiments are only conducted on the FSDD set, but not on the GSC set.
4. Figure 5c. Why does the interval analysis technique perform so much worse on the GSC set relative to the FSDD set?  On a related note, it would also be useful to describe some more details about the model architectures for the two tasks.
5. The section on “Experimental comparison with prior work” similarly left me with a number of questions. The authors mention that “We found that, in practice, optimization approach used by POPQORN produces approximations of slightly smaller volume than our LSTM transformer (although non-comparable).” Could these be quantified and reported in the paper. Also, why are the approximation volumes not comparable between the two systems.

Minor comment: It is true that most works in audio classification and speech recognition use processed frontend features such as MFCCs. However, there is also a significant body of work which operates directly on the time-domain signal. Perhaps it would be better to clarify this in the text?
For example:
Pascual S, Bonafonte A, Serra J. SEGAN: Speech enhancement generative adversarial network. arXiv preprint arXiv:1703.09452. 2017 Mar 28.
Sainath TN, Weiss RJ, Senior A, Wilson KW, Vinyals O. Learning the speech front-end with raw waveform CLDNNs. In Sixteenth Annual Conference of the International Speech Communication Association 2015.


**Experience Assessment:**

I have read many papers in this area.

**Review Assessment: Checking Correctness Of Derivations And Theory:**

I assessed the sensibility of the derivations and theory.

**Review Assessment: Checking Correctness Of Experiments:**

I assessed the sensibility of the experiments.

**Review Assessment: Thoroughness In Paper Reading:**

I read the paper at least twice and used my best judgement in assessing the paper.

---

> ### Author Response · Authors · 2019-11-12
> **Reply for Reviewer#3**
>
> Thank you for the feedback and interest in our work. Below we answer the concerns:
>
> Q1: While I found section 3 to be useful to get an intuition of the proposed method, I still feel that it could be condensed a bit to add in additional details. For example, the authors don’t describe “back-substitution” in the work, which I believe should be described in the main text.
>
> A1: We added more details on back-substitution in the last paragraph of Section 3. We have not described back-substitution in full detail in the main text as it is not part of our main contributions. Full details of this algorithm can be found in [1]. We also provide full derivation of bounds in our overview example using back-substitution in Appendix C.
>
>
> Q2: How sensitive was the provability metric to the choice of these 100 test examples?
>
> A2: To check the sensitivity of our results to the choice of 100 samples that we verified we ran the verification on 10 random permutations of our test set (each time with a different seed). We describe this experiment in Appendix F and show the verification results in Figure 9 with error bars indicating the variance. These results show that provability is not significantly affected by the choice of 100 element subset used for verification. In the next revision, we will include these error bars for each plot.
>
>
> Q3: The section on “Provable defense for audio classifiers” was not very clear to me. The authors state that “To train, we combine standard loss with the worst case loss obtained using interval propagation.” I was not clear on what the modified loss is. Could the authors please clarify this in the text, preferably a mathematical formulation? Also, I’m curious why these experiments are only conducted on the FSDD set, but not on the GSC set.
>
> A3: In Appendix G we have now provided the detailed description of our provable defense procedure, including a mathematical formulation of a modified loss, which closely follows Gowal et al. (2018). We also ran this experiment for GSC network —  these results are also presented in Appendix G.
>
>
> Q4: Why does the interval analysis technique perform so much worse on the GSC set relative to the FSDD set? Could you describe more details about the model architectures used for the two tasks?
>
> A4: Based on our experiments, we believe that certifying GSC is more difficult for two reasons: (i) the main factor which impacts provability is the number of frames — each additional frame causes accumulation of errors introduced by loose convex relaxations which ultimately makes the certification more likely to fail. This reflects in the fact that GSC, where samples have 19 frames on average, is significantly more difficult to certify than FSDD which has samples with 15 frames on average, and (ii) given that GSC and FSDD have 30 and 10 output classes, respectively, in the case of GSC verifier needs to prove that the final logit of correct class is greater than the logit of all other 29 classes which is naturally more difficult than proving it for only 9 other classes as is the case with FSDD.
>
>
> Q5: Could you quantify and report the difference in the volume between POPQORN and your approach? Also, why are the approximation volumes not comparable between the two systems?
>
> A5: Please see our answer to the main points above where we provide answers to these questions and in-depth comparison between POPQORN and our approach.
>
>
> Q6: Could you clarify that there is significant body of work which operates directly on the time domain signal?
>
> A6: Yes, thank you for the helpful references. We made a clarification and added the references you suggested.
>
>
> [1] Singh, Gagandeep, et al. "An abstract domain for certifying neural networks." Proceedings of the ACM on Programming Languages 3.POPL (2019): 41.

---

### Author Response · Authors · 2019-11-12
**Main Points for Common Concerns**

We thank the reviewers for their comments. We first answer the main points, followed by specific questions:

 - Comparison with POPQORN:

  We provide detailed quantitative comparison between our approach and POPQORN in Appendix E. The main takeaways are:

  1) Our method produces bounds strictly better than interval bounds (see Theorem 1). This means the maximum distance between the true function and our bounds cannot grow arbitrarily large. POPQORN offers no such guarantees. Although POPQORN uses gradient descent to optimize for the bounds with minimum volume, there are no convergence or optimality guarantees. This can cause imprecise results in practice — we found many inputs in our audio benchmarks for which POPQORN produces bounds worse than intervals, please see Figure 6 in Appendix E for one example. Our bounds are strictly better than intervals and do not suffer from this problem.

  2) While we experimented with POPQORN on synthetic inputs and found that it indeed produces relaxations with slightly smaller volume than our method, for realistic inputs which appear in our audio benchmarks it often performs worse. As it is non-comparable to intervals, meaning that all points obtained via intervals are not included inside those obtained via POPQORN and vice-versa, its bounds are often worse than intervals, e.g. see Figure 6.

  3) As POPQORN is relatively slow (108 minutes per sample), we evaluated it only on 10 samples. We plugged their relaxation of tanh * sigmoid instead of our bounds and demonstrated that it results in 0 verified samples on our benchmarks. At the same time, we verify 4 out of these 10 samples (in 29 seconds per sample). We believe the core reason is the existence of many pathological cases as described in the previous point where gradient descent used by POPQORN converges to suboptimal solution which ends up being worse than interval bounds. We note that we contacted the authors to confirm that we are using their framework correctly.

  4) POPQORN is 100 - 1000 times slower than DAC as it relies on optimization via gradient descent whereas DAC produces bounds in constant time. This makes POPQORN less practical for verification at the scale of audio classifiers.


 - Speedup of the existing implementation:

  We optimized our implementation which improved the runtime of back-substitution. This results in a significant speedup of end-to-end verification: for update depth of 3 now it takes 17.74 seconds per input, compared to 87.92 seconds reported in Table 2. For larger back-substitution depths, the speedup is even larger. We will update running time analysis for experiments in Table 2.

---

### Decision · Program_Chairs · 2019-12-19

**Decision:**

Reject

**Comment:**

The paper developed log abstract transformer, square abstract transformer and sigmoid-tanh abstract transformer to certifiy robustness of neural network models for audio. The work is interesting but the scope is limited. It presented a neural network certification methods for one particular type of audio classifiers that use MFCC as input features and LSTM as the neural network layers. This thus may have limited interest to the general readers.

The paper targets to present an end-to-end solution to audio classifiers. Investigation on one particular type of audio classifier is far from sufficient. As the reviewers pointed out, there're large literature of work using raw waveform inputs systems. Also there're many state-of-the-art systems are HMM/DNN and attnetion based encoder-decoder models. In terms of neural network models, resent based models, transformer models etc are also important. A more thorough investigation/comparison would greatly enlarge the scope of this paper.